# Prospective Observational Study of a Non-Arthroscopic Autologous Cartilage Micrografting Technology for Knee Osteoarthritis

**DOI:** 10.3390/bioengineering10111294

**Published:** 2023-11-08

**Authors:** Dimitrios Tsoukas, Ilie Muntean, Christos Simos, Ruben Sabido-Vera

**Affiliations:** 1Orthopaedic Clinic for Advanced Arthroscopic Sports and Regenerative Surgery, MITERA General Maternity and Children’s Hospital, 15123 Athens, Greece; christossimos@hotmail.com; 2Hospital of Sant Joan Despi Moises Broggi, 08970 Barcelona, Spain; iusauto@hotmail.com; 3Independent Researcher, 08031 Barcelona, Spain; rubensabidovera@gmail.com

**Keywords:** autologous, intra-articular, knee osteoarthritis, micrograft technology, pain management, regenerative medicine

## Abstract

Autologous micrografting technology (AMT^®^) involves the use of autologous micrografts to stimulate/enhance the repair of damaged tissue. This study assessed the efficacy and safety of the AMT^®^ procedure in patients with early stages of knee osteoarthritis. Briefly, the AMT^®^ procedure involved extraction of auricular cartilage, disaggregation using the Rigeneracons^®^ SRT in 4.0 mL of saline solution, and injection of the disaggregated micrografts into the external femorotibial compartment area of the affected knee. Ten patients (4 men, 6 women; age range: 37–84 years) were included in the study. In all patients, there was a steady improvement in knee instability, pain, swelling, mechanical locking, stair climbing, and squatting at 1- and 6-months post-procedure. Improvement in mobility was observed as early as 3 weeks post-procedure in 2 patients. Significant improvements were seen in mean scores of all five subscales of Knee Injury and Osteoarthritis Outcome Score (KOOS [KOOS symptoms, KOOS pain, KOOS ADL, KOOS sport and recreation, and KOOS quality-of-life]) between pre-procedure and 1- and 6-months post-procedure (all *p* ≤ 0.05). Autologous auricular cartilage micrografts obtained by AMT^®^ procedure (using Rigenera^®^ technology) is an effective and safe protocol in the treatment of early stage knee osteoarthritis. These encouraging findings need to be validated in a larger patient population and in a randomized clinical trial (RCT).

## 1. Introduction

Knee osteoarthritis (OA) is a common progressive joint disease, characterized by chronic pain and functional disability [1]. The pooled global prevalence of knee OA in the year 2020 was 16.0% in individuals above 15 years of age and 22.9% in those aged 40 years and over [2]. The prevalence and incidence of knee OA increases with age and is higher in women than in men [2]. It represents a substantial and increasing health burden with considerable personal, economic, and societal toll [1,3].

OA is characterized by the degradation of articular cartilage and bone matrix components [4]. The aim of treatment is to preserve joints in order to improve pain, restore activity, and delay arthroplasty [5]. Bone marrow stimulation techniques, cartilaginous or chondrogenic tissue repair, osteochondrogenic autologous transplantation, and autologous chondrocyte implantation [5] are the most widely used options for treating OA. In addition, injection of corticosteroids, hyaluronic acid into the intra-articular region, and subchondral injections of platelet-rich plasma are other widely used techniques in OA patients [6]. However, none of the aforementioned treatment options effectively arrest structural deterioration of cartilage and bone or successfully reverse existing structural defects [3,4,7]. They also do not enable regeneration of the articular tissue with its distinct functional characteristics [8]. As OA is a highly heterogeneous disease, targeting a single joint tissue for treatment may not be effective [4].

Regenerative medicine is currently considered a promising therapy for the treatment of OA in humans. This innovative method involves the use of biological sources such as stem cells and grafts that allow the regeneration and replacement of cells, tissues or organs in order to restore the original structure and physiological functions [4,9]. Autologous micrografting technology (AMT^®^) involves the use of autologous micrografts to enhance regeneration of an impaired or damaged tissue. The key strengths of AMT^®^ are a good safety profile, non-rejection of the injected micrografts, and specificity to recover normal functioning of tissues through specific signaling pathways [10]. Use of autologous micrografts can overcome some of the current limitations of other therapeutic approaches, like invasiveness, donor site morbidity, cell death, and allogeneic response [11]. The AMT^®^ procedure is not considered an advanced therapy medicinal product as it falls under the category of non-substantially manipulated cells or tissues used for the same essential function, which means that the cells when removed from their original environment in the human body are used to maintain the original function(s) in the same anatomical or histological environment under allogeneic or autologous conditions [12].

The Rigenera^®^ technology (Regenera Activa Worldwide S.L., Barcelona, Spain, and Human Brain Wave SRL, Torino, Italy) is a novel strategy for tissue mechanical disaggregation, which allows obtaining autologous micrografts [13,14] enriched in progenitor cells expressing MSC-like markers and having strong regenerative potential [15]. The AMT^®^ procedure using Rigenera^®^ technology has been used worldwide for more than ten years since its development in 2012 to stimulate and enhance self-regenerative processes for multiple conditions. It has shown clinical efficacy in the management of complex wounds [16,17,18], for the regeneration of the bone in periodontal surgeries [19], for pinched nose deformity [20,21,22], for improving hair density in patients affected by androgenetic alopecia [23,24,25,26], for treatment of scars [27], for treatment of osteochondral lesions of the knee [28], and for the treatment of knee chondropathy [5]. The procedure has also shown promise for treatment of osteochondral defects in a preclinical study [11].

The aim of this study was to determine the clinical effect and outcomes of the AMT^®^ procedure in patients with early stages of knee OA.

## 2. Materials and Methods

This was a prospective, open-label study conducted between June 2022 to November 2022.

### 2.1. AMT^®^ Procedure

The AMT^®^ procedure used in this study was like the one used in previous studies in knee-joint-degeneration patients [5,24,27]. The tools used in the AMT^®^ procedure are Rigeneracons^®^ SRT (7945RS), Sicurdrill^®^ 2.0, Sicurlid and Sicurstick (Regenera Activa Worldwide S.L., Figure 1a–d).

The Rigeneracons^®^ SRT is a sterile disposable class IIa medical device designed to mechanically disaggregate solid human tissue to obtain soluble autologous micrografts, which are injectable (the AMT^®^ solution). Rigeneracons^®^ SRT comprises a grid with 100 hexagonal holes, each containing 6 calibrated microblades; a helix that rotates through an internal metal ring at a constant 80 revolutions per minute to disaggregate tissues with a cut off of 80 µm without causing cellular disruption and maintaining cell viability; and 3 arms. The Sicurdrill^®^ 2.0 is a class I medical device with a motor that provides the electromechanical impulse to rotate the helix of the Rigeneracons^®^ SRT. The Sicurdrill^®^ 2.0 operates in 1-min cycles, each cycle started by pressing the frontal button. It is specially designed to be easily transportable. The Sicurlid and Sicurstick are specific adapters to secure the Rigeneracons^®^ SRT to the Sicurdrill^®^ 2.0. The procedure is shown in Figure 2 and described below:

Preparation and anesthesia: The auricular concha of the ear was disinfected with chlorhexidine and the area for biopsy extraction was marked with a dermal marker. The area was anesthetized with 2.5 mL of 2% lidocaine without vasoconstrictors; 0.5 mL in the retroauricular nerve to anesthetize and the rest on both sides of the auricular concha to hydro separate the skin from the cartilage. Adrenaline was used if the bleeding did not stop after applying 5 min of hemostatic pressure.

Extraction: Three biopsy punches were made in the marked area with a 2.5 mm dermal punch. The three biopsies were placed on the grid of the Rigeneracons^®^ SRT, and manually rotated to cover them under the helix. The donor area in the ear was covered with a band-aid.

Disaggregation and collection: Approximately 4.0 mL of saline solution was added through the extraction hole of the Rigeneracons^®^ SRT with a Luer Slip syringe, until the biopsies were slightly embedded in the solution. Then the cap was closed, the Sicurstick inserted in the Rigeneracons^®^ SRT and placed into the Sicurlid. The device was then attached to the Sicurdrill^®^ 2.0. The frontal button on the Sicurdrill^®^ was pressed to start the tissue disaggregation, which was indicated by the yellow flashing LED on the Sicurdrill^®^. The Sicurdrill^®^ was operated for six consecutive cycles, each cycle of 1-min duration. At the end of six cycles, 4.0 mL of the AMT^®^ solution was collected with a Luer Slip syringe.

Infiltration: The external femorotibial compartment area of the affected knee was disinfected with chlorhexidine. The 4.0 mL of AMT^®^ solution was injected with a 21G ½, 0.8 × 40 mm needle into the external femorotibial compartment. The joint was mobilized to distribute the injected AMT^®^ solution evenly across the treated area.

All steps were carried out under sterile conditions and the full Rigeneracons^®^ SRT kit which is intended for single use was safely disposed of after use in accordance with local guidelines.

Joint inflammation, if any, in the first 24–72 h post-procedure was treated with analgesics (other than nonsteroidal anti-inflammatory drugs which may interfere with the effectiveness of the micrograft). Patients were advised to rest in positions that do not overstrain the treated joint for at least 7 days post-procedure. To ensure healing of the biopsy site in the ear, patients were advised to not shower or have a sauna bath in the first 24–48 h post-procedure, to change the dressing of the biopsy site after 24 h and to keep the biopsy site clean and dry.

The study was conducted in accordance with the Declaration of Helsinki. Good Manufacturing Practices rules for processing, Good Clinical Practices for the clinical application and ethics committee approval did not apply to the study as it fell under the category of “autologous use in one step surgery, minimal manipulation, monofunctional use (used for the same essential function in the recipient as in the donor), and manipulation with devices in aseptic conditions” [24,25].

All patients provided written informed consent before the start of the procedure.

### 2.2. Eligibility Criteria

Patients included in the study were at the age between 35 and 75 those diagnosed with OA grade 2–3 (except for two patients with OA grade 3–4) on Kellgren and Lawrence scale [29], meaning that the joint preserved at least part of the articular cartilage.

Patients excluded were those with any autoimmune diseases including rheumatoid arthritis; those with any concomitant uncontrolled metabolic diseases; patients undergoing systemic and/or local corticosteroids treatment; hyaluronic acid infiltration within the previous six months; as well as those with grade 4 advanced OA.

### 2.3. Assessments

Magnetic resonance imaging (MRI) and X-ray were performed pre-procedure. The Knee Injury and Osteoarthritis Outcome Score (KOOS) questionnaire [30] recommended by the International Knee Documentation Committee was used for assessing pain, stiffness, and function pre-procedure and at 1- and 6-months post-procedure. The KOOS is a self-reported outcome measure assessing the patient’s opinion about the health, symptoms, and functionality of their knee. It is a 42-item questionnaire, including 5 subscales: symptoms (seven items), pain (nine items), function in daily living (ADLs [17 items]), sports and recreation function (five items), and quality of life (four items). Each item has five possible answer options scored from 0 (No Problems) to 4 (Extreme Problems) and each of the five scores is calculated as the sum of the items included. Scores are transformed to a 0–100 scale, with zero representing extreme knee problems and 100 representing no knee problems. An aggregate score of all subscales is not calculated since it is regarded desirable to analyze and interpret the five dimensions separately [30].

Incidence of adverse events was monitored during the study. The biopsied area in the ear was checked for signs of infection and for any abnormality in healing. A clinical evaluation and safety assessment was carried out at 12 months post-procedure.

### 2.4. Statistical Analysis

Data were analyzed with one-way analysis of variance and posterior multiple comparison Bonferroni test. Data were expressed as the mean ± standard deviation (SD). A *p* ≤ 0.05 was considered statistically significant. RStudio software (RStudio Team (2020). RStudio: Integrated Development for R. RStudio PBC, Boston, MA, USA) was used for statistical analysis.

## 3. Results

The study included 10 patients, 4 men and 6 women; all except one patient were aged 53 years and above (Table 1). Patients reported pain in their affected knee for the past <1 year to 10 years prior to start of study.

### 3.1. MRI and X-ray Findings

MRI at baseline revealed subchondral bone edema, chondropathy patella, and medial compartment narrowing; Baker cyst was noted in one patient (Table 2). X-ray revealed presence of osteophytes and medial compartment narrowing (Table 2).

### 3.2. AMT^®^ Procedure Outcomes

In all patients, the wound in the ear completely healed within one week without any need for stitches. None of the patients needed adrenaline to stop the bleeding from the biopsied area in the ear. There were no cases of infection or signs of anormal healing of the donor area like “cauliflower ear”. The patients strictly followed physician’s aftercare recommendations for the healing of the donor area.

The AMT^®^ procedure was successful in all 10 patients. Pain resolution and good recovery of daily activities was seen in all 10 patients, with improvement in mobility being observed as early as 3 weeks post-procedure in 2 patients (Table 3).

Physical examination at 1- month and 6-months post-AMT^®^ procedure demonstrated a steady improvement in all patients in knee instability, pain, swelling, mechanical locking, stair climbing and squatting.

### 3.3. KOOS Subscale Scores

All 10 patients completed the assessments at 1- and 6-months post-procedure. Significant improvements were seen in the mean scores of all five subscales of KOOS (KOOS symptoms, KOOS pain, KOOS ADL, KOOS sport and recreation, and KOOS quality-of-life) between pre-procedure and 1- and 6-month post-procedure (Figure 3, all *p* ≤ 0.05). Between 1- and 6-months post-procedure, significant improvements were observed in the mean scores of the subscales of KOOS symptoms, KOOS pain, and KOOS ADL (all *p* ≤ 0.05), while for the subscales of KOOS sport and recreation, and KOOS quality-of-life, the improvements observed at 1-month were maintained at 6-months. Each one of the KOOS subscales was analyzed and passed the Bonferroni test.

### 3.4. Safety and Long-Term Follow-Up

Clinical evaluation at 12 months post-procedure showed that all 10 patients had continued good mobility and minimal or no pain in the affected knee.

The AMT^®^ procedure showed a good tolerability profile since none of the patients reported any adverse event during the study.

## 4. Discussion

Use of biological strategies such as endogenous stem cells or tissue-specific progenitor cells to enable patient’s own cartilage regeneration ability represents an important advance in the treatment of cartilage defects and OA [31,32].

Chondrocytes, which are highly specialized cells in the articular cartilage of the knee constitute approximately 1% of total volume [33] and supply the extracellular matrix (ECM) with the elements such as collagen and proteoglycans that constitute the cartilage function [34]. Chondrocytes are responsible for replacement of ECM molecules lost through degradation over the years. However, aging decreases the ability of chondrocytes to replace ECM molecules resulting in progressive degeneration of articular cartilage leading to joint pain and dysfunction, clinically identified as OA [35]. Also, while articular cartilage can withstand intensive and repetitive physical stress, impact forces caused by falls, sports injuries and road accidents lead to substantial damage [8], but the articular cartilage has limited ability to repair and self-renew as it is an avascular tissue [8,36,37]. Autologous chondrocytes are an effective cell therapy source for repair of chondral defects, but the challenges of using them are the paucity of donor sites, de-differentiation during expansion in culture and early apoptosis [38]. MSCs are another treatment option with a potential for cartilage regeneration but their use is limited by their tendency to differentiate into various cell lineages and the need for specific induction manipulation to promote chondrogenesis [38].

Perichondrial tissue, which is present in all adult cartilage tissues except for articular and fibrocartilage [39], has the potential to produce hyaline-like cartilage [40]. The clinical relevance of perichondrium was recognized more than a century ago [39], and it is possibly among the first tissues to attract clinical interest for its regenerative potential, with initial reports on its role in cartilage regeneration made as early as the 19th century [40,41,42]. Interest in the perichondrium was revived in recent years owing to its possible role as a microenvironment containing stem and chondrogenic progenitor cells [38,39]. Findings from multiple animal and human studies both in vivo and in vitro indicate that perichondrium can facilitate and is essential for cartilage regeneration and has the clinical potential to treat various cartilage-associated diseases and traumas [39]. Perichondrium contains a pool of mesenchymal progenitor cells called chondroblasts. Chondroblasts contribute to tissue homeostasis by replicating and differentiating into chondrocytes. An in vitro study showed that the progenitor cells in perichondrium had paracrine effects on prolonging the lifespan and promoting the proliferation and cartilaginous expression of chondrocytes [38]. Subcutaneous implantation of progenitor cells and chondrocytes together in nude mice was found to promote formation of cartilaginous tissue [38].

Auricular cartilage as a source of perichondrial progenitor cells has the advantages of ease of harvest and minimal donor-site morbidity compared with other sources of cartilage with perichondrium like the nasal cartilage [38]. Animal studies have demonstrated that autologous micrografts generated from auricular cartilage were positive for cartilage progenitor cell markers such as CD44, CD90, and CD117, and the ribonucleic acid derived from micrografts was positive for tissue cartilage markers such as Sox-9 and COL2A1 [22]. In comparison with murine MSCs, these levels indicate presence of chondrocyte-progenitor cells with a mesenchymal phenotype in auricular micrografts [22]. Autologous micrografts from human auricular cartilage cultured in vitro with human chondrocytes were found to positively influence chondrocyte differentiation as shown by both increased glycosaminoglycans deposition and the presence of collagen II and stimulate secretion of cartilage trophic factors such as insulin-like growth factor 1 and transforming growth factor β, without affecting chondrocyte viability [31]. The principle behind AMT^®^ procedure with auricular cartilage for knee OA is to transfer the signaling potential of chondroprogenitor cells, chondrocytes and signaling factors together with ECM components to enhance the restoration of joint function and reduce pain and stiffness [5].

In this study, use of AMT^®^ procedure with auricular micrografts was found to be effective and safe in the treatment of early stage and moderate knee OA. Post-procedure, all patients reported minimal or no pain. Improvement in mobility was observed as early as 3 weeks post-procedure, which was maintained even after 12 months post-procedure, indicating early benefits maintained over the long-term. These observations were supported by the significant improvements observed in all five subscales of KOOS, a patient-reported outcome (PRO) measure, at 1- and 6-months post-procedure. PROs are increasingly recognized by regulators, clinicians, and patients as valuable tools to collect patient-centered data [43,44]. The KOOS scale, an extension of the Western Ontario and McMaster Universities Arthritis Index (WOMAC) is mainly focused on the knee and is better adapted for younger, more active populations [43]. It has adequate internal consistency, test-retest reliability and construct validity in young and old adults with knee injuries and/or OA [45]. A change of 8–10 points is suggested to represent the minimal perceptible clinical improvement of the KOOS [43]. In this study, the change in KOOS subscale scores from pre-procedure ranged between 20 and 30 points at 6-month post-procedure.

The findings from this study in patients with early to moderate stage of knee OA strengthen the observation from in vitro and animal studies on the use of autologous auricular micrografts for joint pain management. The observations from this study and that of Marcarelli et al. [5] suggest a potential role for AMT^®^ procedure in the treatment of various articular cartilage defects as well as cartilage defects in other joints in the body. Future studies in larger patient populations are needed to validate such clinical application. Limitations of the study are the small sample size, absence of a control group, lack of long-term follow-up and findings being based on a single AMT^®^ procedure.

## 5. Conclusions

Autologous auricular cartilage micrografts obtained by AMT^®^ procedure (using Rigenera^®^ technology) is an effective and safe protocol in the treatment of early stage knee OA and could be a valid treatment approach. The findings need to be validated in a larger patient population and in a randomized clinical trial (RCT).

## Figures and Tables

**Figure 1 bioengineering-10-01294-f001:**
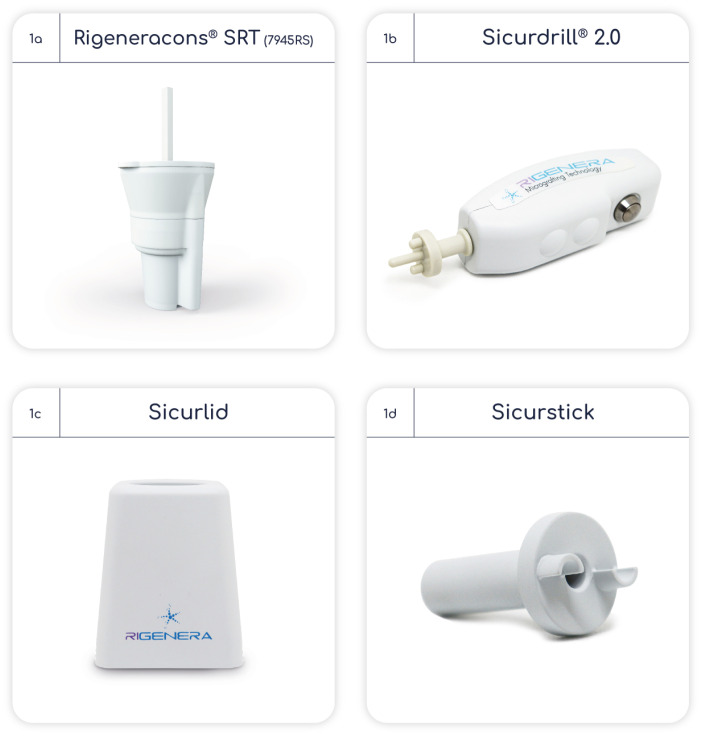
Tools used in autologous micrografting procedure.

**Figure 2 bioengineering-10-01294-f002:**
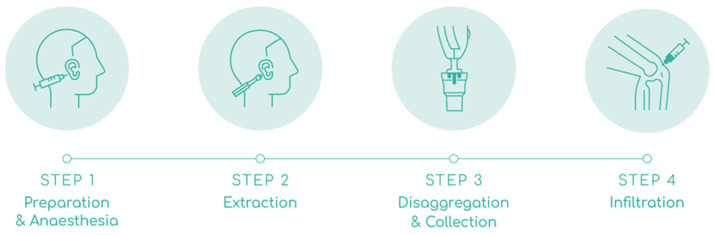
Steps involved in autologous micrografting procedure. The figure shows AMT^®^ procedure in 4 steps. STEP 1: preparation and anesthesia of donor’s area. STEP 2: extraction of 3 auricular cartilage biopsies. STEP 3: mechanical disaggregation of the biopsies and collection of AMT^®^ solution. STEP 4: infiltration of AMT^®^ solution into the knee joint.

**Figure 3 bioengineering-10-01294-f003:**
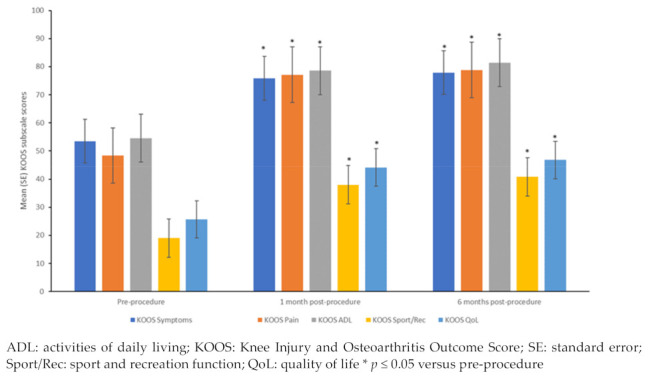
KOOS subscale scores pre-procedure and at 1- and 6-months post-procedure.

**Table 1 bioengineering-10-01294-t001:** Patients’ baseline demographics and disease characteristics.

Patient Number	Age (Years)	Sex	BMI (kg/m^2^)	Diagnosis Based on KL Scale	Clinical History	Past Medications	Past Procedures
1	63	Male	25	OA grade 3; chondropathy in left knee	Pain for the past 4 years and stiffness which became worse around a month before enrollment into study. Medial meniscus posterior horn tear 10 years prior to study enrollment which was not treated	None	None
2	84	Female	23	OA grade 3–4	Pain in right knee from past 10 years	None	None
3	64	Female	22	OA grade 3–4	Left knee pain for 8 years, worsening over time. Gonarthrosis for 8 years	NSAIDs	PRP injections in 2020 and 2021
4	37	Male	25	Chondropathy grade 3	Gonarthrosis	NSAIDs and paracetamol	Physiotherapy and PRP
5	74	Female	23	OA grade 3	Gonarthrosis. Pain in affected right knee for the past 10 years that became worse after an accident nearly 7 months prior to study enrolment	None	None
6	73	Female	22	OA grade 2	Pain for past 2 years	NSAIDs for 3 months	Physiotherapy
7	73	Female	22	OA grade 1-early 2	Very mild pain for the past less than 1 year	None	None
8	64	Female	23	OA grade 1 early 2	Pain for the past one year. Fibromyalgia and depression	None	None
9	53	Male	31	OA grade 1 early 2	Pain, reduced mobility and grinding sensation (crackling)	NSAIDs, education about weight control	Physiotherapy
10	53	Male	31	OA grade 3 symptomatic	Mild pain and reduced mobility	None	None

BMI: body mass index; KL: Kellgren–Lawrence; NSAIDs: nonsteroidal anti-inflammatory agents; OA: osteoarthritis; PRP: platelet-rich plasma therapy. Note: Patient number 4 was an ex-professional basketball player; Patient number 5 was a very active person, bicycling for 40 years, but had stopped it about a year before enrollment into study.

**Table 2 bioengineering-10-01294-t002:** Magnetic resonance imaging and X-ray findings pre-procedure.

Patient Number	MRI Findings	X-Ray Findings
1	Subchondral bone edema medial femoral condyle with small cartilage lesion, medial meniscus degenerative tear. Chondropathy patella grade III.	Standing X-ray: medial compartment narrowing
2	Chondropathy patella II–III and medial femoral condyle grade III–IV.	Osteoarthritis KL scale grade 3–early 4
3	Medial compartment narrowing with chondropathy both condyles.Baker cyst 4 cm × 9 cm	Osteophytes present; KL scale grade 3 to 4 of osteoarthritis with medial compartment narrowing
4	Patella alta 3rd degree chondropathy patella and cartilage wear 1.7 cm × 1.5 cm at the femoral trochlea	Patella alta
5	Chondropathy with diffuse cartilage defects	Osteoarthritis grade 3
6	Chondropathy patella grade III with cartilage defects	Osteoarthritis grade 2
7	Diffuse cartilage wear Outerbridge grade II	Osteoarthritis grade 1–2
8	Three-plane study of the left knee. Internal meniscus of preserved morphology and signal. External meniscus with increased horizontal signal that also extended to the lower articular surface compatible with meniscal tear and associated with the presence of small loculated cystic images that extended anteriorly with a larger component that crossed the lateral retinaculum which might correspond to the palpable swelling, compatible with parameniscal cyst. Conclusion: Rupture of the external meniscus body with parameniscal cyst extending anteriorly, crossing the lateral retinaculum. Slight decrease in the thickness of the patellar cartilage in its superior and central portion.	Osteoarthritis KL grade 1–2
9	Chondropathy wear lateral tibial condyle	Osteoarthritis of the femorotibial joint left knee KL grade 1–2
10	Cartilage wear medial femoral condyle Outerbridge grade 2 and medial tibial condyle grade 2–3.	Osteoarthritis KL grade 3

KL: Kellgren–Lawrence; MRI: magnetic resonance imaging; NA: not available.

**Table 3 bioengineering-10-01294-t003:** Clinical outcomes within 12 months following AMT^®^ procedure.

Patient Number	Description of Clinical Condition
1	No pain and good mobility at 10 months post-procedure
2	Lateral pain and good mobility as early as 3 weeks post-procedure
3	Less pain and better mobility as early as 3 weeks post-procedure
4	Impressive improvement in mobility and pain reduction from within 2 months post-procedure. The patient started bicycling post-procedure, which was stopped earlier due to pain
5	Improvement in clinical outcomes as early as 6 weeks post-procedure The patient started running at 2 months and playing basketball at low intensity at 3 months
6	Able to do daily activities without any pain. Clinical improvement observed was maintained even after 11 months post-procedure
7	No pain while carrying out daily activities, while pre-procedure the patient could not be on foot for more than 2 h The patient was able to walk for longer periods and was also able to take the stairs instead of the elevator Clinical improvement observed was maintained even after 1-year post-procedure
8	No pain and good mobility which was maintained even after 10 months post-procedure
9	Reduced pain and ability to walk long distances of more than 10 km with only minimal pain The patient lost some weight, but not sufficient to significantly help with the overall body inflammation and biomechanics in the joint
10	Reduced pain and improved mobility

AMT: autologous micrografting technology.

## Data Availability

Data are contained within the article.

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
