# Peer review of "Prospective Observational Study of a Non-Arthroscopic Autologous Cartilage Micrografting Technology for Knee Osteoarthritis"

_bioengineering, 2023, doi:10.3390/bioengineering10111294_

Round 1
Reviewer 1 Report
Comments and Suggestions for Authors
Tsoukas et al.'s article covers a topic that falls within the scope of the journal. The article is both timely
and pertinent. Several aspects should be improved by the authors in order to publish it:
· Please delete “encouraging” from the title. Add to the title the type of study: “prospective observational study”
· Line 69. “Discovery”? Better “developed” …
· What is the particle size of the micrografts?
· Line 124: Infiltration. Please describe the infiltrated volume. Perhaps 4 mL?
· Lines 164-165. Please add a reference to this statement.
· There are some discrepancies between table 1 and table 2, specifically in the OA grade. Please unify it. For example, patient 2 has OA 2-3 in table 1 and the same patient has 3-4 in table 2.
· Table 3. Please describe the following time of the Clinical outcomes.
· Figure 3. As a suggestion, I this that is more informative to group the graph in function of Koos domain and not in function of the following time. That is, all blue bars together… all orange together,…
· Lines 273-277. These are the postulated mechanisms of action of the micrograhps. Please explain if cells are integrated in the native cartilage. Please explain if the matrix (fragmented) is integrated in the cartilage. If no, what happens with the cartilage fragments?
· Please consider changing the CONCLUSIONS and the abstract: “The findings need to be validated in a larger patient population.” This is true, but incomplete. The results must be validated in randomized clinical trials (RCT).
Reference # 8, please check the style. Check all references for Bioengineering style. Check also reference 33.
Comments on the Quality of English LanguageMinor editing of English language required
Author Response
Dear Mr./Mrs.
Thank you for reviewing the manuscript.
Please, see the attachment for further details on your comments.
Kind regards,
Dr Dimitrios Tsoukas.

Reviewer 2 Report
Comments and Suggestions for Authors
The authors provide preliminary insights into the effectiveness and safety of Autologous Micrografting Technology (AMT®) for early-stage knee osteoarthritis (OA), using autologous micrografts sourced from the auricular cartilage. They present results from a 6-month follow-up involving 10 patients of various ages and OA severity.
Regrettably, I must point out that this study lacks essential research methodology elements, including criteria for patient selection, sample size, outcome measures, and the inclusion of control groups. The data is derived from a very small and highly diverse patient group with a short follow-up period, rendering it inadequate for evaluating treatment effectiveness or safety. I strongly recommend expanding the sample size and, if possible, conducting a randomized controlled trial (RCT), with potential registration prior to commencing the study, utilizing a conventional infiltrative osteoarthritis treatment like hyaluronic acid or corticosteroids.
Comments on the Quality of English Languageenglish is fine. only minor revisons required
Author Response

(The authors gave the same response as above.)

Reviewer 3 Report
Comments and Suggestions for Authors
Remarks to the Author:
In this article, the authors assessed the efficacy and safety of the AMT® procedure in patients with early stages of knee osteoarthritis.There are some suggestions listed as follows:
1. Characters in the figure 1 should be larger.
2. Is there a physical or surgical examination for the prognosis criteria ?
3. The author should show specific data about clinical evaluation at 12 months post-procedure.
4. Authors could consider more supporting reference, which are related to osteoarthritis and exosomes in the introdcution, such as “Biomater Transl. 2023, 4 (1): 27-40. ” “Biomaterials translational 2023, 4(2): 67-84.” “Journal of orthopaedic translation, 37, 69–77.” “Journal of orthopaedic translation, 38, 141–155.”
Comments on the Quality of English LanguageExtensive editing of English language
Author Response

(The authors gave the same response as above.)

Round 2
Reviewer 1 Report
Comments and Suggestions for Authors
All concerns have been addressed.
Comments on the Quality of English LanguageAll concerns have been addressed.
Author Response
Thank you for the revision.
Regards,
Dr. Dimitrios Tsoukas.
Reviewer 2 Report
Comments and Suggestions for Authors
The authors acknowledge the main study limitations but do not adequately address the raised concerns. Injecting auricular cartilage micrografts into the knees of ten patients shows no significant advantage over the common and cost-effective corticosteroid injection or even a placebo saline injection, as demonstrated in a recent systematic review (Previtali et al. Cartilage. 2021 Dec;13(1_suppl):185S-196S. doi:10.1177/1947603520906597). The study, at best, highlights the treatment's safety and technical feasibility, but fails to establish its cost-effectiveness and actual benefits as an alternative for knee OA therapy. I recommend that the authors resubmit the article with an increased number of cases and longer follow-up periods, at the very least
Author Response
Thank you for the revision.
Regards,
Dr Dimitrios Tsoukas.
Reviewer 3 Report
Comments and Suggestions for Authors
The authors have addressed the comments. I recommend to accept the paper.
Comments on the Quality of English LanguageExtensive editing of English language.
Author Response

(The authors gave the same response as above.)
